

# Hydroclimatic processes as the primary drivers of the Early Khvalynian transgression of the Caspian Sea: new developments

Alexander Gelfan [1,2], Andrey Panin [1,3], Andrey Kalugin[1], Polina Morozova [3],

Vladimir Semenov [1,3,4], Alexey Sidorchuk [2], Vadim Ukraintsev [1,3], Konstantin Ushakov [1,5]

[1] Water Problems Institute, Russian Academy of Sciences, Moscow, 119333, Russia
[2] Lomonosov Moscow State University, Faculty of Geography, Moscow, 119991, Russia
[3] Institute of Geography, Russian Academy of Sciences, Moscow, 119017, Russia
[4] Obukhov Institute of Atmospheric Physics, Russian Academy of Sciences, 119017 Moscow, Russia
[5] Shirshov Institute of Oceanology, Russian Academy of Sciences, 117997 Moscow, Russia

*Correspondence to:* Alexander Gelfan (hydrowpi@mail.ru)

**Abstract.** It has been well established that during the late Quaternary, the Khvalynian transgression of the Caspian Sea occurred, when the sea level rose tens of meters above the present one. Here, we evaluate the physical feasibility of the hypothesis that the maximum phase of this extraordinary event (known as the "Early Khvalynian transgression") could be initiated and maintained for several thousand years solely by hydroclimatic factors. The hypothesis is based on recent studies dating the highest sea level stage (well above +10 m a.s.l.) to the final period of deglaciation, 17-13 kyr BP, and studies estimating the contribution of the glacial waters in the sea level rise for this period as negligible. To evaluate the hypothesis put forward, we first applied the coupled ocean and sea-ice general circulation model driven by the climate model and estimated the equilibrium water inflow (irrespective of its origin) sufficient to maintain the sea level at the well-dated marks of the Early Khvalynian transgression as 400-470 km$^3$/year. Secondly, we conducted an extensive 14C-dating of the large paleochannels (signs of high flow of atmospheric origin) located in the Volga basin and found that the period of their origin (17.5-14 ka BP) is almost identical to the recent dating of the main phase of the Early Khvalynian transgression. Water flow that could form these palaeochannels was earlier estimated for the ancient Volga River as 420 km$^3$/year, i.e. close to the equilibrium runoff we determined. Thirdly, we applied a hydrological model forced by paleoclimate data to reveal physically consistent mechanisms of an extraordinarily high water inflow into the Caspian Sea in the absence of visible glacial meltwater effect. We found that the inflow could be caused by the spread of post-glacial permafrost in the Volga paleo-catchment. The numerical experiments demonstrated that the permafrost resulted in a sharp drop in infiltration into the frozen ground and reduced evaporation, which all together generated the Volga runoff during the Oldest Dryas, 17-14.8 kyr BP, up to 360 km$^3$/year (i.e. the total inflow into the Caspian Sea could reach 450 km$^3$/year). The closeness of the estimates of river inflow into the sea, obtained by three independent methods, in combination with the previously obtained results, gave us reason to conclude that the hypothesis put forward is physically consistent.

## 1 Introduction

Paleogeographical data give grounds to assert that during the late Quaternary the largest highstand in the Quaternary history of the Caspian Sea took place, which was called the "Great" Khvalynian transgression. The boundaries of the Khvalynian Sea are well-detected in the relief of the Northern Caspian lowland (e.g. Leontiev, 1968, 1977; Rychagov, 1974, 1997), and confirmed by stratigraphic and biostratigraphic analysis of Quaternary deposits (Fedorov, 1957, 1978; Svitoch and Yanina 1997; Svitoch, 2009, 2014; Yanina 2012; Makshaev and Svitoch 2016; Yanina et al., 2018;



Kurbanov et al., 2021). The accumulated data show that in the early, maximum stage of the Khvalynian transgression,
the sea level rose up to +48 m a.s.l., i.e. almost 80 meters above the current Caspian Sea level (CSL), while the sea
surface area was 940,000 km$^2$, which is 2.5 times larger than its current area (Yanko-Hombach and Kislov, 2018).
Although the very fact of the Early Khvalynian transgression and the assessment of the maximum sea level are not
questioned by most researchers, there are significant disagreements regarding the dating of this extraordinary
hydrological phenomenon and the views on its genesis.
In the 1970s-90s, it was assumed that the maximum phase of the Khvalynian transgression was synchronous to the
Early Valdai (Early Weichselian, MIS 4) glaciation of the Russian Plain and occurred 50-70 ka BP (see reviews by
Kislov et al., 2014; Arslanov et al., 2016 and references there). Accumulation of geochronometric, mostly radiocarbon
(14C) data has allowed a reassessment of this viewpoint and proposal for a younger age of the Early Khvalynian
transgression, corresponding to the second half of the last glaciation (Late Valdai, Late Weichselian, MIS 2) (Svitoch
et al., 1994, 1998; Svitoch and Yanina, 1997). A number of compilations of the accumulated geochronological data
have been published in recent years that enable a more detailed interpretation of the transgression. Arslanov et al.
(2016) summarized the 14C and 230Th/234U dates of the Lower Khvalynian deposits performed at St. Petersburg
University and proposed to date the +35 and +22 m a.s.l. transgressive stages at 16 and 14 ka BP, respectively, while
the period 14-12 ka BP was attributed stages 0 and -12 m a.s.l. of the subsequent Late Khvalynian transgression.
Krijgsman et al. (2019), based on a review of available dates, assigned the entire Khvalynian epoch to the 35-10 ka
BP interval, with the Yenotaevian regression separating the Early and Late Khvalynian phases, about 15 ka BP.
Koriche et al. (2022) attributed the Early Khvalynian stage to 35-25 ka BP and the Late Khvalynian stage to 17-12 ka
BP. The latter, according to (Koriche et al., 2022), reached +35 m a.s.l. during 14.5-16.5 ka BP. Makshaev and Tkach
(2023), based on generalization of more than 180 14C dates, attributed the Early Khvalynian stage of the Caspian Sea
to the period 46-12.5 ka BP. In their opinion, sea level exceeded the contemporary level at the beginning of MIS 2
(28-25 ka BP). This was followed by two transgressive events of 25-18 ka BP (level reached +10+15 m a.s.l.) and 17-
13.5 ka BP (+20÷+22 m a.s.l.), separated by a regressive phase between 18 and 17 ka BP. These authors attribute the
Yenotaevian regression and the subsequent Late Khvalynian transgression to 12.5-8.5 ka BP.
Recently, a series of papers have been published where sections containing the Khvalynian sediments were first dated
by optically stimulated luminescence (OSL) (Kurbanov et al., 2021, 2022, 2023; Butuzova et al., 2022; Taratunina et
al., 2022). These results were summarized in Kurbanov et al. (2023), who identified the following transgression stages:
1) sea level rise to about +5 m a.s.l (32 m above the present CSL) between 30-35 and 27 ka BP; 2) sea level stabilization
with a slight (about 2 m) rise within the interval of 27-20 ka BP; 3) a sharp rise in the sea level beginning from 18-17
ka BP; 4) maximum stage of the sea level during the period around 16-15 ka BP; 5) rapid fall of the sea level during
the period 15-14 ka BP from its maximum values to less than +11 m a.s.l.
Thus, the Khvalynian stage in the development of the Caspian Sea can currently be referred to the period from the end
of MIS 3 (about 35 ka BP) to the Early Holocene (8.5 ka BP). At the beginning of that period, the sea level was lower
than it is now, but no later than 27 ka BP it was already much higher. It should be emphasized that no direct dates for
the maximum stage of +48+50 m a.s.l. have been obtained in any study. The recently published OSL data on the
Raygorod section in the Northern Caspian Lowland at +13.5 m a.s.l. (Taratunina et al., 2022) show that from at least
90 ka BP up to 18 ka BP, subaerial deposits (alluvium, loess) were accumulating there, i.e., the maximum phase of
transgression could not have occurred before the Last Glacial Maximum (LGM). The age of the maximum stage is
best justified by (Kurbanov et al., 2023), where the maximum stage is sandwiched between the rise and fall phases



and is assigned to the interval of 15-16 ka BP. Therefore, taking into account the reliable recent dating reviewed above,
we will limit our attempt to explain the genesis of the Early Khvalynian transgression to the final period of
deglaciation, (18)17-13 kyr BP.
Another widely debated question is: what are the causes of the Early Khvalynian transgression? The discussed
hypotheses are reduced to the consideration of the sources of a huge water influx into the sea, which, under the climatic
conditions of the Late Pleistocene, could provide the sea level rise of tens of meters above the present CSL. Other
causes, such as tectonic factors or natural, internal fluctuations of the water body, are considered unlikely (Rychagov,
1997; Yanko-Hombach and Kislov, 2018, respectively).  According to paleoclimatic modeling experiments (e.g.
Kislov and Toropov, 2007; Morozova, 2014; Yanko-Hombach and Kislov, 2018; Morozova et al., 2021), the LGM
and post-LGM climate is characterized by low air temperatures and low precipitation with a reduced, relative to the
modern, climatic runoff, that is, the difference between precipitation and evaporation in the catchment area of the
Caspian Sea. To explain the increased river inflow into the Caspian Sea as a factor of the Early Khvalynian
transgression, hypotheses are put forward about additional, in comparison with atmospheric precipitation, sources of
water. The most discussed hypothesis is the recharge of glacial meltwater from the south-eastern flank of the
Scandinavian ice sheet (SIS) via the Volga River during the LGM and deglaciation (Kvasov, 1979; Varuschenko et
al., 1987; Toropov and Morozova, 2011; Tudryn et al., 2016; Koriche et al., 2022). Hypotheses are also put forward
about the overflow of glacially dammed lakes and water discharge from outside the drainage basin of the Caspian Sea
- from the upper Dnieper catchment and from the Sukhona and Vychegda Rivers that belong to the Arctic Ocean
catchment  (Kvasov, 1979; Larsen et al., 2006; Lyså et al., 2011), from the Aral Sea basin through a hypothetical
hydrological system connecting it with both the ice-dammed lakes of the West Siberian ice-sheet and the Caspian Sea
(see Grosswald and Kotlyakov, 1989; Chepalyga, 2007, as well as a critique of this hypothesis by Svitoch (2009) and
Panin et al. (2020)). Kvasov (1979) estimated the contribution of the SIS meltwater and proglacial lakes as 46% and
input from the Aral Sea as 21% of the total water inflow into the Early Khvalynian Caspian Sea, which was estimated
by this author as 560 km3/year. Based on the PMIP2 (Paleoclimate Modelling Intercomparison Project, Phase 2)
climate simulation data, Toropov and Morozova (2011) estimated that the SIS meltwater could have made the main
contribution to the Khvalynian transgressions: 83% of the ancient Volga River inflow assessed as 462 km3/year. The
coupled atmosphere-ocean-vegetation HadCM3 climate model experiments allowed Koriche et al. (2022) to conclude
that meltwater combined with the changes (due to isostatic adjustment) in the drainage system leading to an increase
in the Caspian Sea catchment area by 60-70% of its modern size, had the most  substantial influence on the sea level
rise during the last deglaciation from 20 kyr BP to 14 kyr BP. Note that all the above estimates of the SIS meltwater
contribution were obtained solely from modelling results, which were not confirmed by geological and/or
geomorphological evidence.
The validity of the above hypotheses considering glacial meltwater as a substantial source of water inflow into the
Caspian Sea and confidence in the corresponding estimates of meltwater contribution to the Early Khvalynian
transgression, are directly related to the assessed age of the transgression. According to the present-day state of
geochronological studies described above, the stages well above +10 m a.s.l. are dated to the period of (18)17-13 kyr
BP. Tudryn et al. (2016) proposed that glacial meltwater entered the Caspian Sea during the entire deglaciation epoch
up to 13.8 kyr BP. However, Panin et al. (2021) showed that the inflow of meltwater into the Volga basin occurred
only from its upper part directly covered by the Scandinavian ice-sheet, and was limited to a period from 21 to 16.5
kyr BP, i.e. the transgression was developing towards its highest stage, while the input of glacial waters ceased. The
authors estimated the possible glacial meltwater input to the upper Volga River in the range of 15-70 km$^3$/year, or



only 5–25% of the present-day Volga runoff into the Caspian Sea, which is far from enough to support the Khvalynian
highstand. The insignificant role of glacial meltwater in the genesis of the Early Khvalynian transgression during the
deglaciation period is also argued in earlier works (Kalinin et al, 1966; Panin et al., 2005; Sidorchuk et al., 2009).
Also, a number of recent studies (Panin et al., 2020, 2022; Borisova et al., 2022) showed that neither the proglacial
lakes in the upper Volga region proposed by Kvasov (1979), nor the overflow to the Volga River from the Arctic basin
occurred in MIS 2.
The hypothesis of hydroclimatic initiation of the Early Khvalynian transgression, in the absence of a noticeable
contribution from glacial meltwater, is supported by the ubiquitous presence in the southern half of the Eastern
European Plain, including the Volga basin, of signs of high flow of atmospheric origin - river palaeochannels that are
many times greater in size than the contemporary rivers (Sidorchuk et al., 2009, 2011, 2021; Ukraintsev, 2022). On
the basis of the developed morphometric analysis of palaeochannels, Sidorchuk et al. (2009, 2021) estimated the
meteoric (formed due to atmospheric precipitation) runoff of the ancient Volga River, which was capable of forming
the palaeochannels, as 420 km³/year, i.e. 65% higher than the modern annual runoff. At physically reasonable ratios
of precipitation and evaporation in the Caspian Sea, this is quite sufficient to maintain levels of the Early Khvalynian
transgression (Sidorchuk et al., 2009; Kislov et al., 2014).
The age of large palaeochannels in the Dnieper, Don, and Volga basins obtained by the 14C method falls within the
interval of 18-13 kyr BP (Borisova et al., 2006; Sidorchuk et al., 2009; Panin et al., 2013, 2017; Panin and Matlakhova,
2015), that is, exactly at the time when the CSL rose above +10 m a.s.l. However, it should be noted that in the Volga
basin itself, only two large palaeochannels have been dated so far on the Moskva River, a tributary of the Oka River,
and on the Samara River, a tributary of the lower Volga (Sidorchuk et al., 2009). This is insufficient for such a large
basin encompassing several natural zones with significant differences in the present climate. In this study, we clarified
the period of activity of large palaeochannels in the Volga basin.
Thus, according to the above review there is a knowledge gap, which drives the main motivation for our study. On
the one hand, the well-founded modern datings show that in the final period of deglaciation, 18(17)-13 kyr BP, the
CSL rose well above +10 m a.s.l. (likely, up to +22 ÷ +35 m a.s.l.), but, on the other hand, it has been proved that the
meltwater runoff – due to the Scandinavian ice-sheet melting and outbursts of ice-dammed proglacial lakes  - was
either absent or contributed insignificantly to the transgression of the sea during this period. A research question arises:
could the Early Khvalynian transgression of the Caspian Sea have been initiated and maintained solely by
hydroclimatic factors in the cryoarid climate of the deglaciation period and in the absence of an inflow of glacial
meltwater?
Kislov and Toropov (2007), Sidorchuk et al. (2009) hypothesized that during the decline in the glacier melt, river flow
into the sea could significantly exceed the current one due to the spread of post-glacial permafrost in the river
catchments of the East European Plain. Permafrost could reduce evaporation for the sea catchment territory owing to
a drastic decrease in the infiltration capacity of frozen ground. Gelfan and Kalugin (2021) applied a physically based
hydrological model to assess the sensitivity of the Volga River runoff to the hypothetical spread of permafrost in the
river basin. The authors demonstrated that under the modern climatic conditions mean annual runoff may increase by
85% due to modeled "freezing" of the basin. They concluded that river inflow into the Caspian Sea is markedly
sensitive to presence of permafrost over the sea catchment area, thus further verification of the hypothesis is advisable
in the cryoarid climatic conditions of the late Pleistocene. One of the objectives of our study is to verify this hypothesis
explaining the maintenance of the CSL at +22 ÷ +35 m a.s.l. reliably dated to the period of 18(17)-13 kyr BP in the
absence of significant glacial meltwater runoff during this period.



The logic of our study was as follows. Using a full ocean model coupled with a model of sea-ice dynamics INMIO
COMPASS – CICE  (Ibrayev et al., 2012; Hunke et al., 2015), we simulated the Caspian Sea water balance
components under the climate conditions of the Late Pleistocene – Middle Holocene, which were re-constructed with
the help of the climate model INMCM4.8 (Volodin et al., 2018). On the basis of the simulation data, we estimated the
equilibrium river water inflow into the sea maintaining its level at the well-dated marks of the Early Khvalynian
transgression. To verify the model-based estimations, the river runoff assessments derived from the morphometry of
palaeochannels formed in the period 18-13 kyr BP (Sidorchuk et al., 2021) were used. Also, we made an attempt to
improve the knowledge on the chronology of widespread geomorphological evidence of high river runoff in the Late
Pleniglacial – Late Glacial in the Volga basin. To achieve this, additional dating of large palaeochannels in different
parts of the basin was carried out. Then, the hydrological model was forced by the paleoclimate data, and numerical
experiments were conducted to assess the water inflow to the Caspian Sea from the ancient Volga catchment with
underlying permafrost. Comparison of estimates of water inflow into the Caspian Sea obtained using three independent
approaches (1 – estimating equilibrium inflow into the sea via an ocean model coupled with a climate model; 2 -
paleogeographic reconstructions of water flow through palaeochannels, and 3 – hydrological modeling river runoff
generation in the sea catchment area under the paleoclimatic conditions) provided us with grounds for answering the
above research question.
The remaining part of this paper is organized as follows. General information about the Caspian Sea is given in the
next section. Section 3 contains methodology of our study including brief description of the models used and the
numerical experiments designed. The results are presented and discussed in Section 4. The overall conclusions are
given in Section 5.
**2. General information on the Caspian Sea**
The Caspian Sea (36°33'–47°07' N, 46°43'–54°50' E) is the world's largest inland water body located within an
endorheic (no outflow) basin. The sea surface area at the current sea level is equal to 365,000 km$^2$. The coastline
length is 5970 km. The greatest length of the sea (along the meridian 50°00'E) is 1030 km. The greatest width along
the parallel 45°30' N reaches 435 km. The large meridional extent results in climate variations over the basin: from
sub-tropical in the southwest to desertic in the east and northeast.
Owing to the endorheic nature of the Caspian Sea, its level widely fluctuated in the past. During the late Cenozoic,
the CSL variations exceeded, probably, several hundreds of meters (Forte and Cowgill 2013) and at least 100 m,
during the last 500,000–700,000 years (Water balance…, 2016), during the Holocene the CSL changes were from 15
m (Water balance…, 2016) to several tens of meters (Kakroodi et al. 2012), during the last millennium the CSL
changed by 10 m (Naderi Beni et al. 2013) and during the period of instrumental observations (beginning from 1830)
within the range of 4 m: from -25.1 m a.s.l. at the beginning of 1880s to -29.0 m a.s.l. in the middle of 1970s (Frolov,
2003). The present (December of 2022) CSL is -28.6 m a.s.l.
The CSL variations are controlled mainly by water inflow from rivers and precipitation on the sea, as well as by water
outflow through evaporation from the sea surface (Ratkovich, 1993; Golitsyn et al., 1998; Kroonenberg et al., 2000;
Arpe and Leroy, 2007; Arpe et al., 2012; Naderi Beni et al., 2013; Panin and Dianskii, 2014; Chen et al., 2017), i.e.
they are strongly dependent on climatic variations (Kroonenberg et al., 2000; Arpe an Leroy, 2007; ), at least as long
as no significant changes are occurring in the sea catchment area. Groundwater inflow contribution is estimated to be
small (Zektser, 1996) and expected to partly compensate for the impact from the outflow to the Kara-Bogaz-Gol Bay
(Chen et al., 2017) accounting for the uncertainty of both estimates.



The Caspian Sea is fed by more than 130 large and small rivers with the total annual flow of about 300 km$^3$ (average
value for 1880-2001 (Frolov, 2003)). The total catchment area of the sea is 3,050,000 km$^2$, which is 8 times the area
of its water area (386,400 km$^2$ at the sea level of -27.50 m a.s.l.). The largest of the tributaries is the Volga River,
whose catchment area is 1,360,000 km$^2$. For the period of instrumental observations (1881-2012), the mean annual
flow of the Volga in the river outlet (Volgograd city) is about 250 km$^3$ (e.g. Arpe et al., 2019). Taking into account
water losses due to evaporation in the Volga delta, the Volga water inflow into the Caspian Sea is about 233 km$^3$ of
water per year (Frolov, 2003) or about 80% of the total inflow of river water into the sea.  According to (Kislov and
Toropov, 2007), the relative contribution of the Volga runoff has changed insignificantly over the past 20 thousand
years and accounts for 75 to 90% of the total inflow into the Caspian Sea. According to various estimates, the long-
term mean precipitation on the Caspian Sea surface in the 20th century was about 200 mm/year (about 77 km$^3$/year),
evaporation from the sea surface was 960 mm/year (about 371 km$^3$/year), and effective evaporation (the difference
between evaporation and precipitation) was 760 mm/year (about 294 km$^3$/year), respectively (e.g. Frolov, 2003; Water
Balance…, 2016).
The relationship between water input to and output from the Caspian Sea controls the sea level. The CSL response to
changes in the main water balance components of the sea depends on the peculiarities of the sea bathymetry, namely,
a significant fraction of shallow water areas. The northern part of the sea is shallow, in the southern and central parts
of the sea there are deep depressions that are intersected by an underwater ridge. The average depth of the sea is 208
m, the maximum depth is 1025 m. About 69% of the total sea area is at depths less than 200 meters, and a shallow
zone with depths less than 10 m occupies 28% of the sea area. In the range of the CSL fluctuations from -28.0 to -
24.0 m a.s.l., a one-meter change in the CSL results in a 1500 km$^2$ change in the area of the deep-water part of the sea,
and a 12500 km$^2$ change in the area of the shallow-water North Caspian part (Frolov, 2021). The predominant increase
in the water area due to the shallow waters of the Northern Caspian with a rise in the sea level creates a non-linear
dependence of evaporation from sea level fluctuations (Frolov, 2003).

**3 Research Methods**
**3.1 Hydro- thermodynamics model of the Caspian Sea**
To simulate the Caspian Sea water balance components, we used a regional configuration of the coupled ocean and
sea-ice general circulation model INMIO COMPASS – CICE. This approach involves a detailed description of marine
dynamic processes with a high spatiotemporal resolution, taking into account ice drift and energy-mass transfer in the
water-ice-atmosphere system. Thus, it is possible to obtain more reasonable values of evaporation from the sea surface
compared to global climate models, in which a coarser resolution is typically used and the sea level is set constant,
allowing no change in the surface area when the water balance of the sea is different from zero. The importance of
using a full ocean model for the Caspian Sea was demonstrated by (Arpe et al., 2019).
The coupled model built from INMIO COMPASS (Ibrayev et al., 2012) and CICE (Hunke et al., 2015) codes in the
CMF2.0 software environment (Kalmykov et al., 2018) was used earlier for weather forecasting and climate research
(Fadeev et al., 2018; Kalnitskii et al., 2020; Ushakov and Ibrayev, 2018, and references therein). The model solves
the equations of three-dimensional dynamics and thermodynamics of the ocean and sea ice cover, explicitly
reproducing a wide range of processes responsible for the main energy-carrying elements of the circulation. The

 

calculations were performed using a model configuration tuned for the Caspian Sea region with a spatial resolution of
about 22 km and a time step of 20 minutes, which was described in (Morozova et al., 2021).

**3.2 Assessing equilibrium river inflow into the paleo-Caspian Sea under the transgressive levels of the sea**

To assess an equilibrium river inflow into the paleo-Caspian Sea, the paleo-climate data simulated by the INMCM4.8
climate model (Volodin et al., 2018) were set as atmospheric boundary conditions for the coupled ocean-ice model
according to the protocols of PMIP4 (Paleoclimate Modelling Intercomparison Project, Phase 4) and CMIP6 (Coupled
Model Intercomparison Project, Phase 6). The paleo-climate data represent two periods: the Last Glacial Maximum
(experiment LGM, 21 kyr BP, Kageyma et al., 2021) and the mid-Holocene (experiment midHolocene, 6 kyr BP,
Brierley et al., 2020). The data included near-surface air temperature and specific humidity, precipitation, wind
velocity vector, fluxes of incoming longwave and shortwave radiation. The time resolution of the boundary fields was
6 hours, which made it possible to explicitly consider a wide range of variability, from synoptic to interannual scales.
Since the Caspian Sea in the experiments of the climate model was specified in the modern coastline, the isolines of
some boundary fields (air temperature and humidity, incoming longwave radiation) showed a tendency to follow this
coastline. For these fields, an extrapolation was made from the sea area domain adopted by the climate model to the
area of transgression. Since the sea level rise affects mainly the northern coastal regions, the extrapolation was
performed from south to north using the meridional gradients calculated for each field by the least square method over
the central part of the climate model water area (Fig. 1a).

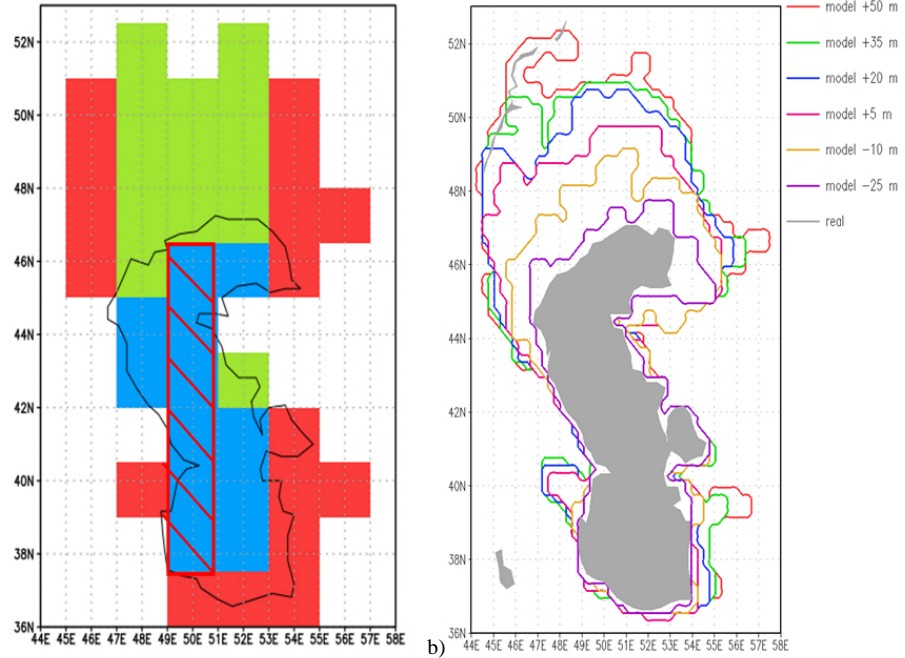

a) b)

**Figure 1: a) The Caspian Sea area representation in the climate model INMCM4.8 (blue cells), red shading - cells used to
calculate meridional gradients, green and red cells - extrapolation areas for transgressive stages (green cells – meridional
extrapolation, red cells – extrapolation by the nearest neighbor method); b) The model representation of the Caspian Sea
coastline for the sea levels assigned in the numerical experiments. The grey fill shows modern boundaries of the sea.**



Further, for several transgressive cells, where this meridional procedure is not applicable, a simple extrapolation by
the nearest neighbor method was performed. Precipitation, wind velocity components, and incoming shortwave
radiation were used directly without extrapolation.
Calculations of the water balance in the LGM and mid-Holocene were carried out for a range of the CSL: from the
near-modern one (-25 m a.s.l.) to the maximum level of the Early Khvalynian transgression (+50 m a.s.l.), with a step
of 15 meters, a total of six experiments. The corresponding model domains are shown in Fig. 1b.
For each of the two paleo-periods and each sea level, the experiment was performed for 50 model years and was
organized as follows (Table 1). First, a rough initial approximation for the annual mean river runoff was specified as
a linear function of the sea area (Morozova et al., 2021). After that, a model spin-up was performed for five years, and
then during the next 15 years of model integration the average water imbalance was calculated. At the end of the 20$^{th}$
year, the obtained average imbalance was subtracted from the river runoff, and the average anomaly was subtracted
from the sea level field. This resulted in the equilibrium runoff value and reinitialized sea level, which were used to
further proceed with the calculations. Another spin-up was performed for 10 years, and finally, the last 20 years of the
experiment were used to analyze the fields of evaporation and precipitation over the sea.

**Table 1 – Stages of numerical experiments with the coupled ocean-ice model**

| Years | Experiment stage |
| --- | --- |
| 1 – 5 | Initial approximation for the runoff. Spin-up. |
| 6 – 20 | Initial approximation for the runoff. Calculating water imbalance. |
| end of year 20 | Applying corrections to runoff and sea level |
| 21 – 30 | Corrected runoff. Spin-up. |
| 31 – 50 | Corrected runoff. Analyzing the Caspian Sea water balance components |

**3.3 Investigating the chronology of large palaeochannels**
Dating was carried out by the radiocarbon (14C) method in the laboratories of the Institute of Earth Sciences, St.
Petersburg University (index LU) and the Institute of Geography, Russian Academy of Sciences, Moscow (index
IGRAN). Plant remains and dispersed organic matter in gyttja were used for dating. Fresh water mollusk shells, which
are frequently met in drill cores, were not used because of the high probability of date distortion due to the hard water
effect. Boring for organics sampling was carried out by a mechanical corer, usually in the centre of the palaeochannel
(depending on its accessibility for the machine). The geological structure of the palaeochannels usually distinguishes
3-4 sedimentary units, from top to bottom: (1) overbank alluvia - silty loam, sandy loam, or peat in place of the filled
up oxbow lake; (2) oxbow lake sediments - clayey loam; (3) sediments of the intermediated stage of the palaeochannel
abandonment, when it was not yet completely isolated from the river and flow still continued; usually silty sand or
sandy silts; (4) channel alluvium - sands, sands with gravel and pebbles. Below the bed of channel alluvium
corresponding to the studied palaeochannel, there were often older alluvial deposits, which could be of diverse
composition - sands, loams, gyttja (unit 5).
Samples from channel alluvium (unit 4) are preferred for dating as they correspond to the time of active palaeochannel
development. However, the channel alluvium is well-washed and organic inclusions are rare. They are much more
commonly found in unit 3 sediments. The process of gradual abandonment of channel meanders usually takes a few
years, at the most a few decades. This is less than the usual interval of uncertainty of 14C dates and from the point of



view of geological time can be considered as a moment. Therefore, we considered that the samples from unit 4 also belong to the time of active development of the palaeochannels, its very end. Unfortunately, in unit 4, as well as in unit 5, organic materials suitable for dating were found only in a small number of boreholes. They were much more common in unit 2. Because the existence of an oxbow lake in the palaeochannels could be very long (millennia), samples were taken only from the very bottom of unit 3 and when interpreting the dates obtained, it was taken into consideration that they refer to the time when the active development of the palaeochannels ceased. In addition, in some cases, it was possible to sample at 14C from unit 5, the ancient alluvium underlying the channel alluvium of the palaeochannel under study. Such dates were interpreted as predating the time of activity of the studied palaeochannel.

Thus, in terms of the stratigraphic position, the dates have been divided into three groups:

- dates from units 3, 4, giving the time of activity of large palaeochannels - activity dates;
- dates from unit 2, referring to the time when the studied palaeochannels had already been abandoned - post-dates;
- dates from unit 5, indicating the time when the large palaeochannels were not yet active - pre-dates.

In order to determine the total activity interval of large palaeochannels in the Volga basin within each of the groups, the dates were summarised. For this purpose, the OxCal 4.4 software Sum module (Bronk Ramsey, 2009) was used.

**3.4 Modeling water inflow into the Caspian Sea from the ancient Volga catchment covered by permafrost**

Numerical experiments were carried out with a physically based model of runoff generation in the Volga River basin (Motovilov, 2016; Kalugin, 2022) developed on the basis of the ECOMAG hydrological modeling platform (Motovilov et al., 1999). Earlier, Gelfan and Kalugin (2021) applied the ECOMAG-based model of the Volga basin for assessing the river runoff sensitivity to the hypothetical permafrost distribution over the basin area.

The model describes spatially variable processes of snow accumulation and snowmelt, heat and water transfer within the vegetation-soil system, evapotranspiration, infiltration into frozen and unfrozen soil, soil freezing and thawing, surface, subsurface and groundwater flow into the river network, and river channel flow with a daily time-step. The model inputs include spatially distributed daily precipitation, air temperature and air humidity data. The Volga River basin was schematized onto grid cells with a mean area of 1750 km$^2$.

A detailed description of the ECOMAG-based Volga River model, methods for setting the parameters and model verification results for the modern climate were presented by Gelfan and Kalugin (2021). In particular, it was shown that the developed model is robust against climate changes, i.e. it allows one to obtain stable (in statistical sense) results of hydrological simulations within the Volga River basin for years with contrasting climatic conditions. We consider the robustness of the hydrological model as a necessary condition for its applicability for paleohydrological reconstructions.

As the boundary conditions in our experiments, we used climate data simulated by the MPI-ESM-CR global climate model, which reproduced climate conditions of the deglaciation period (26-0 kyr BP) with prescribed ice sheets and surface topographies from ICE-6G reconstruction (Peltier et al., 2015) within the framework of PMIP4 experiment (Kapsch et al., 2021). The used climate data included monthly series of the near ground meteorological data obtained within a transit experiment Ice6G_P2 (Kapsch et al., 2021) for the last 26,000 years with a hundred-year averaging period. The MPI-ESM-CR model has a spatial resolution of 3.75º in longitude and 3.7º in latitude on average.



For hydrological modeling, we applied climate simulation data for the four following periods: the post-LGM (18-17.1
kyr BP), the Oldest Dryas (17-14.8 kyr BP), the Bølling (14.7-14.1 kyr BP) and the Allerød (14-12.8 kyr BP). Since
a hydrological model requires daily data, the monthly MPI-ESM-CR-simulated data were transformed into the series
of the corresponding daily values by the delta-change temporary downscaling method (Gelfan et al., 2017). For the
transformation, we used daily data of the meteorological observations for the period of 1985-2014 at 306
meteorological stations located within the Volga River basin. As a result, we constructed 30-year artificial time-series
of daily precipitation, air temperature and air humidity, so that their mean values were equal to the corresponding
long-term means calculated from monthly series for each of the four considered paleo-periods. The constructed series
were assigned as the boundary conditions for the hydrological model.
Taking into account that the climatic boundaries of permafrost follow approximately with an isotherm of the mean
annual air temperature below -5° C (Smith, Riseborough, 2002), in our experiments, the presence of permafrost was
assumed if the climatic data demonstrated a drop in the mean annual air temperature in the Volga basin below -5° C,
i.e. by about 10°C less than the mean air temperature in the modern climate (+4.5°C). For all elements of the
computational domain underlain by permafrost, the initial temperature of soils was set as negative from the ground
surface to the depth of 3 meter (the depth of attenuation of the seasonal temperature fluctuations).
The hydrological model also took into account the features of the vegetation cover in the considered paleoperiods.
Simakova (2008) and Makshaev (2019) showed that during the post-LGM and the Oldest Dryas, periglacial tundra
landscapes were common in the ancient Volga basin. The model parameters corresponding to these landscapes were
set using the Global Land Cover Characterization database (Loveland et al., 2000).
**4. Results and Discussion**
**4.1 Estimates of equilibrium river runoff to the Caspian Sea at the Early Khvalynyan transgression levels**
The numerical simulations with the INMIO COMPASS - CICE model (Sec. 3.2) provided estimates of the Caspian
Sea water balance components for a wide range of possible CSLs under climatic conditions of the Last Glacial
Maximum and the Holocene Climatic Optimum. Fig. 2 shows the average simulated values of evaporation and
precipitation (mm/year) over the Caspian Sea surface area, as well as the river runoff volume ($km^3$/year) required to
maintain different prescribed values of the Caspian Sea at equilibrium conditions. As can be seen from Fig. 2, the
average evaporation decreases when the CSL rises. This is related to the peculiarities of the Caspian Sea morphology:
under the CSL rise, the coastline expands predominantly in the northern direction, where temperatures are lower, and
the sea ice cover period is longer. Precipitation, on the contrary, slightly increases, but this growth does not compensate
for the decrease in evaporation, so the average values of effective evaporation for the entire Caspian Sea surface area
also decrease with the rising sea level above -25 m a.s.l. In general, the change in the equilibrium runoff is proportional
to the change in the Caspian Sea surface area, but this dependence is not linear. For the CSL above -25 m a.s.l., the
Caspian Sea expands to the northern flat shore and the increase in the sea area accelerates.
This is accompanied by a decrease in the river discharge increment per unit area increase. For the level range of -25
÷ -10 m a.s.l., this increment is 0.55 $km^3$/year per $10^3$ $km^2$ for mid-Holocene conditions, and 0.40 $km^3$/year per $10^3$
$km^2$ for LGM. For the transgressive +35 ÷ +50 m a.s.l. range, however, it becomes 0.25 $km^3$/year per $10^3$ $km^2$ for both
mid-Holocene and LGM. Under LGM conditions, both evaporation and precipitation over the sea surface area are
much lower than the corresponding values during mid-Holocene. Simulated evaporation is on average 180-200



mm/year lower, and precipitation is 70-90 mm/year lower, which results in 15-20% lower values of the equilibrium
runoff in LGM compared to mid-Holocene conditions for the CSLs above -25 m a.s.l.

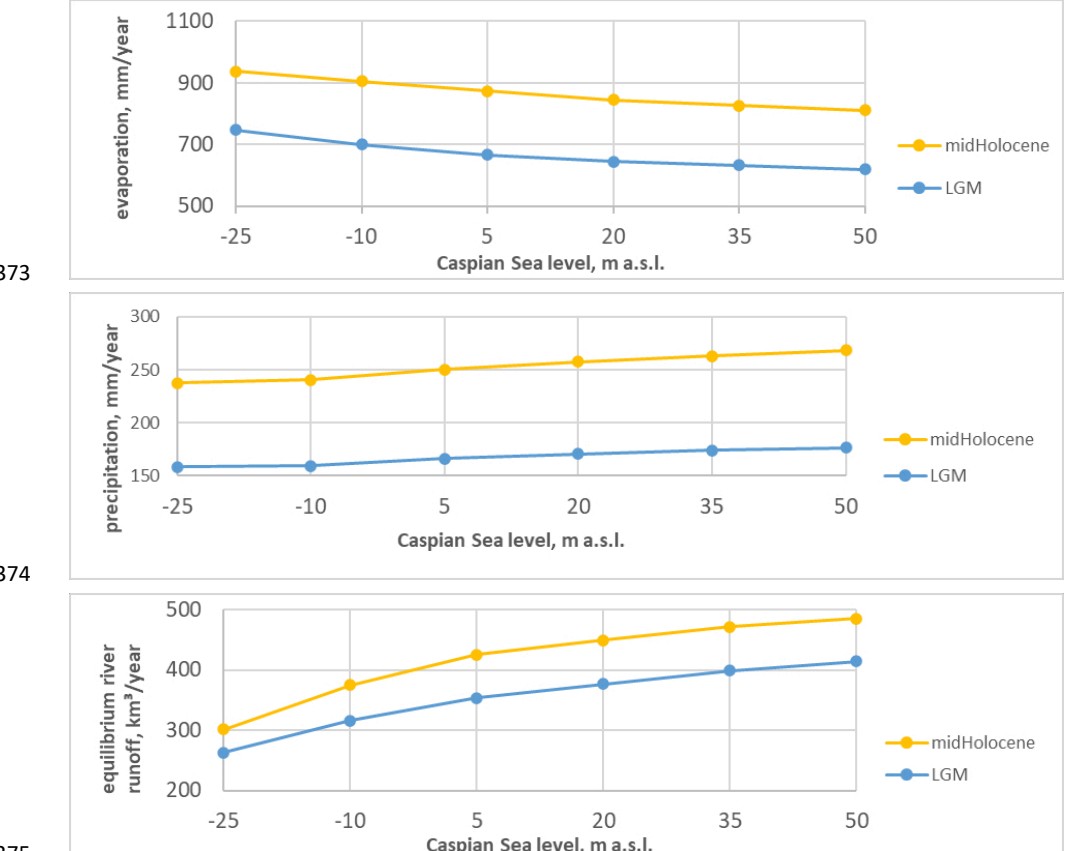




**Figure 2: Simulated Caspian Sea water balance components for different transgressive states under climatic conditions of**
**the Last Glacial Maximum and the Holocene Climatic Optimum:  averaged over the sea area evaporation (a), precipitation**
**(b), and equilibrium river runoff (c) as a function of the sea level.**

Given lower air temperatures during LGM and a large shallow water area in the north at transgressive states of the
Caspian Sea, the sea ice cover extent and duration play a major role in the decrease in evaporation from the sea surface.
Model simulations suggest that the evaporation changes are affected by sea ice export to the warmer southern part of
the Sea driven by sea circulation and surface winds. This effect is important not only during the spring melting season,
but also in winter on the marginal freezing part of the water area, where the sea ice is thin.
The chosen LGM and mid-Holocene periods presumably represent the most contrasting climatic conditions during
the late Pleistocene-early Holocene, so we interpreted the simulated values of the equilibrium river runoff as a possible
range of changes during the deglaciation period under consideration. According to our results, the river runoff values
required to sustain the CSL at the highest dated transgressive state at +35 m a.s.l. (17-13 kyr BP) belong to the range
of 400-470 km³/year. Assuming that the contribution of the Volga River runoff to the total river discharge in that
period was close to the modern one (about 80%), we estimated the river runoff from in the Volga watershed during



the period of the Early Khvalynian transgression ((18)17-13 kyr BP) as 320-375 km³/year, i.e. 1.3-1.5 times larger
than the present day's values.

### 4.2 Results of dating large palaeochannels in the Volga basin

Drilling of large palaeochannels in different parts of the Volga basin was carried out and 14C dates were obtained for
a part of the boreholes (Fig. 3). A total of 57 dates suitable for statistical analysis of the palaeochannel activity time
were obtained. Dates were received from the valleys of 18 rivers: Dubna, Medveditsa, Ustya (upper Volga basin),
Moskva, Protva, Moksha (Oka basin), upper Kama, Izh, Kilmez, Lolog, Yazva, Dema (Kama basin), Samara, Sok,
Buzuluk, B. Cheremshan, B. Kinel (lower Volga basin), B. Uzen (Northern Pre-Caspian).

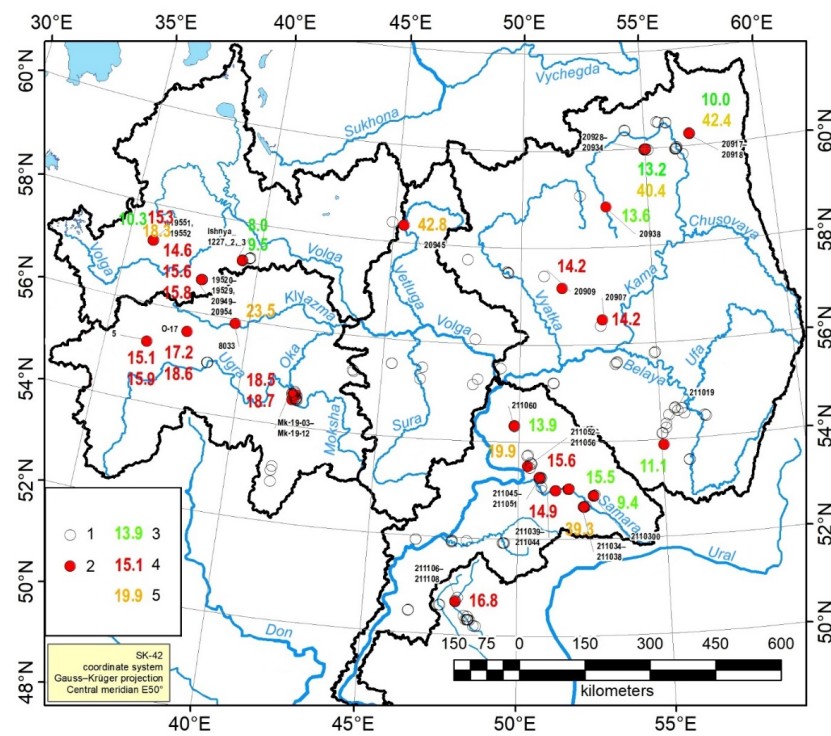


**Figure 3: Map of cores made in large palaeochannels over the Volga basin (1 – all cores, 2 – dated cores; type of dates: 3 – post-dates, 4 – activity dates, 5 – pre-dates. Numbers are central points of 14C calibrated dates).**

All dates are divided into three groups - 19 activity dates, 21 post-dates and 17 pre-dates (see the Methods section)
and for each group the summation was done in OxCal 4.4 (Fig. 4). The resulting distributions suggest the following.
The direct dates in the channel alluvium of the large palaeochannels form two clusters, the main one between 13.8-
17.3 ka BP and a small complementary one between 18.2-18.8 ka BP. The latter overlaps with the youngest part of
the distribution of dates in the underlying sediments (pre-dates), from which we can conclude that, with a high
probability, there is no generation of palaeochannels of the corresponding age. This cluster of dates may be related to
the dating of redeposited ancient organics.



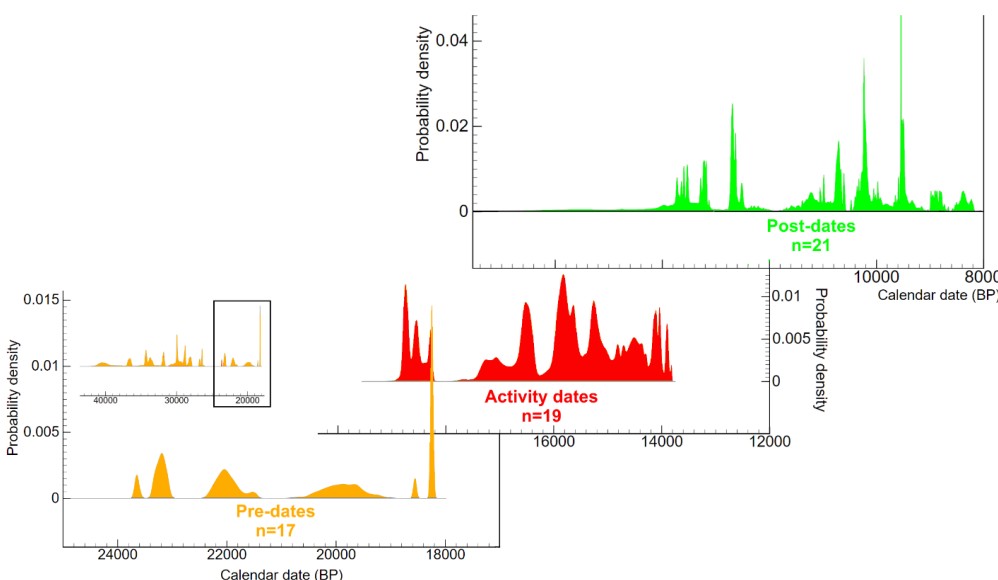

**Figure 4: Summed distributions of radiocarbon dates from large palaeochannels in river valleys of the Volga basin.**

On the right, the distribution of dates in the fluvial alluvium is clearly limited to dates in the overlying sediments (post-dates). It should be noted that in the interval of 12.5-13.8 ka BP, the dates for the overlying sediments are derived from the bottoms of the palaeochannel fills (those cases where there was no material suitable for dating in the channel alluvium). However, at present one can only say with certainty that the stage of large palaeochannel formation and the corresponding epoch of high river runoff in the Volga basin lasted from at least 17.5 to 14 ka BP. A visual analysis of the map (Fig. 3) shows no regional differences in dates, i.e. the epoch started and ended geologically simultaneously in the whole Volga basin. Attention is drawn to the gap in the dates in the interval from 12.5 to 11.5 ka BP corresponding to the Younger Dryas epoch and the very beginning of the Holocene. This may be a result of a shortage of organic material due to scarcity of vegetation during this harsh epoch, but more likely reflects low fluvial activity and a significant drop in river flow in general.

The determined interval of activity of big palaechannels shows that from at least 17.5 to 14 ka BP the Volga River runoff considerably exceeded the modern one. This corresponds generally to the palaeoclimate estimates from paleofloristic data by Borisova (2021) who established a significant increase in atmospheric precipitation in the central East European Plain in the second half of MIS 2 during the warming events 17–19 ka BP (the Late Pleniglacial) and 13–14.5 ka BP (the Bølling and Allerød interstadials). The Oldest Dryas cooling at 14.5–17 ka BP was characterized by a decrease in precipitation below the present-day values, but the high runoff coefficients due to the existence of permafrost could have favored still high runoff values. These estimates point that during the aforementioned period of big palaeochannel activity, the flow hardly remained constant, but it cannot be determined by geomorphological methods: among large palaeochannels there are no distinctive age generations that would differ consistently in size. All large palaeochannels make up a single set of forms, clearly differing in size and position in the valley floor topography from younger palaeochannels, the sizes of which correspond to modern rivers. The distribution of dates for the large palaeochannels also does not reveal clear periodicity or discontinuity on the basis of which the internal periodicity of the high flow epoch could be judged. Perhaps the available number of dates is not yet sufficient for this.



At this stage we can only mark the time frames of the epoch of high river discharge, which began no later than 17.5
ka BP and ended no earlier than 14 ka BP, and relate the estimate of the annual Volga runoff magnitude obtained from
the size of the palaeochannels (420 km$^3$ (Sidorchuk et al., 2021)) to this epoch as a whole. Probably the drop of activity
dates at around 16 ka (Fig. 4) marks the Oldest Dryas pause in high river flow and big channel formation, but to
establish it reliably a much larger massif of dates is necessary.
The interval of increased inflow of river water into the Caspian Sea from 17.5 to 14 ka BP corresponds exactly to the
main phase of the Early Khvalynian transgression dated by marine sediments in the Northern Caspian Lowland from
18-17 to 14-13 ka BP (see the review in the Introduction). It was shown in section 4.1 that such amount of the Volga
runoff was more than enough to keep the Caspian level at +35 m a.s.l. - the highest dated shoreline of the Khvalynian
transgression (remember that the considered maximum level of +48 ÷ +50 m a.s.l. has not yet been characterized by
any direct date - see the review in the Introduction).
What could be the reasons for such a significant increase in river runoff? The involvement of glacial meltwater is
excluded because large palaeochannels are present in various parts of the Volga basin, including those completely
isolated not only from the last, but also from all Quaternary glaciations in general (for example, basins of the lower
Volga or right tributaries of the Oka). It is easy to show that possible increase in river runoff due to thawing of
permafrost, which undoubtedly took place after the LGM, was also negligible. Let us assume that water exchange
between groundwater and river water covered the upper 100 m of the Earth's crust. Let us also assume that during the
last glacial epoch, this entire stratum had a deliberately overestimated ice content of 50%, and the deliberately
unfeasible condition that all meltwater entered the river network when the permafrost melted. It is not difficult to
calculate that if this 100-meter layer of permafrost had melted during the above 3,000-year period, it would have
increased the annual river runoff from the modern basin area by less than 23 km$^3$, which is less than 10% of the
average modern flow volume in the Volga basin. It should be emphasized that this estimate is repeatedly
overestimated. In reality, the additional inflow of water due to melting permafrost could be an order of magnitude
less.
Thus, huge water flowing into the Caspian Sea from the Volga basin during the period from 17 to 13 ka BP could only
be of atmospheric origin (except for possible minor glacial meltwater runoff from the sources of the Volga itself at
the very beginning of this period as demonstrated by Panin et al. (2021)). As mentioned in the Introduction, Gelfan
and Kalugin (2021) quantified a significant decrease in runoff losses due to the hypothetical spread of permanently
frozen soils over the Volga catchment and the resulting increase in the runoff coefficient, i.e. proportion of
precipitation involved in the river runoff formation. But the question arises: is the amount of precipitation
corresponding to the cryoarid climate of the deglaciation epoch enough to form an extraordinary river runoff even
with the spread of permafrost over the catchment area of the Caspian Sea? To answer this question, we carried out
numerical experiments with a hydrological model that reproduce the formation of river inflow into the Caspian Sea in
the climatic conditions of the period from 17 to 13 ka BP and under the assumption of frozen catchment area of the
sea. The results are presented in the next section.
**4.3 Modeling the Volga River runoff in the climate conditions from the post-LGM to the Allerød (18-13 kyr**
**BP)**
Fig. 5 illustrates changes in the mean annual precipitation, air temperature and air humidity deficit assessed from the
MPI-ESM-CR-simulated monthly data and averaged over the Volga basin for four periods: the post-LGM (18-17.1
kyr BP), the Oldest Dryas(17-14.8 kyr BP), the Bølling (14.7-14.1 kyr BP) and the Allerød (14-12.8 kyr BP), covering



the epoch of the Early Khvalynian transgression. According to these data, all considered periods were colder than the
modern climate in the Volga River basin, herewith each subsequent period was warmer than the previous one. Mean
annual precipitation values assessed for different periods were 18-34% less than the modern value. Due to the cold
climate, all the periods are characterized by an increase in the mean annual solid precipitation from 7% in the post-
LGM and the Bølling to 41% in the Allerød (relative to the modern values). On the contrary, the mean annual liquid
precipitation sum decreased from 45% in the Oldest Dryas to 54% in the Bølling. The mean annual air humidity
deficit, which affects evaporation from the catchment surface, turned out to be lower than the modern one by an
average of 40-50% in different periods.

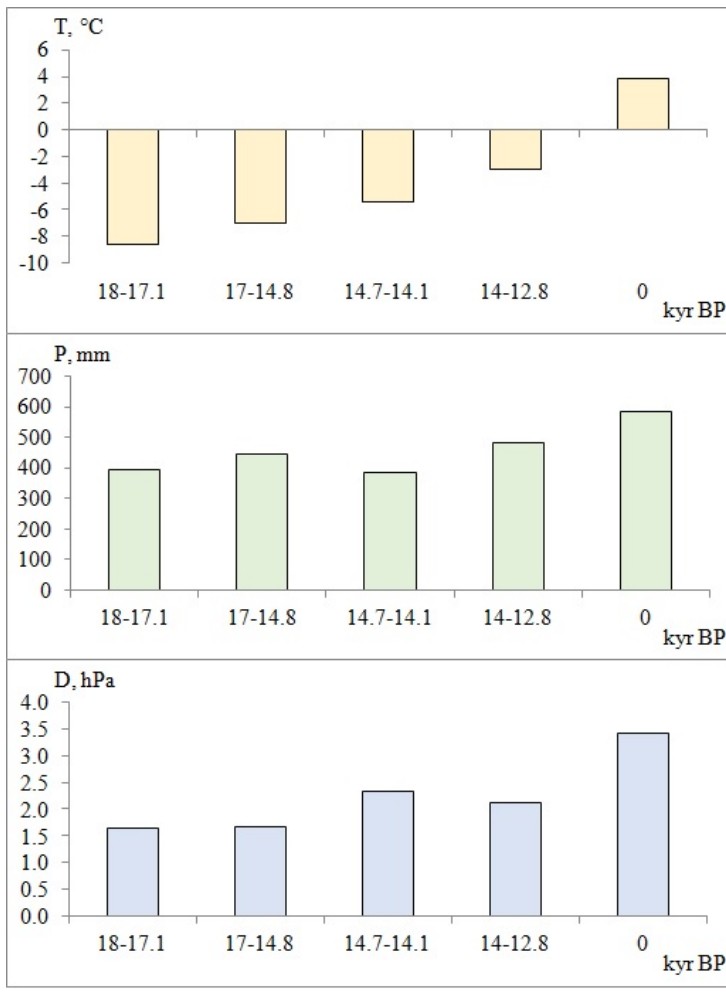

**Figure 5: MPI-ESM-CR-simulated data of the mean annual air temperature, total precipitation and air humidity deficit,**
**averaged over the Volga basin, during the considered periods of paleo-time and under the modern climate.**
Taking into account the cold climate in the post-LGM period, when the average annual temperature was 12.6°C lower
than the present one (see Fig. 5), the Oldest Dryas (10.9°C lower) and the Bølling (9.4°C lower), we assumed that the
whole catchment area was covered by continuous permafrost during these three periods. Generally, this assumption
corresponds to the paleogeographic findings of Sidorchuk et al. (2008) and Borisova (2021). An algorithm that allows



taking into account the hypothetical presence of permanently frozen ground in the Volga River catchment and modeling the hydrological effect of permafrost was described by Gelfan and Kalugin (2021).

Numerical experiments with the hydrological model, which was forced by the temporary downscaled paleo-climate data, demonstrated that the mean annual runoff of the ancient Volga during the post-LGM period and the Oldest Dryas increased in comparison with the modern one for the period of 1985-2014 (259 km$^3$) by 24% and 38%, respectively (Fig. 6). The runoff rising during the Oldest Dryas was larger due to larger mean precipitation. The permafrost led to a decrease in the infiltration capacity of the soils by more than an order of magnitude in comparison with the unfrozen soil over the river catchment. Decreased soil infiltration resulted in an increase in the mean runoff coefficient to as much as 0.67, i.e. 2/3 of precipitation falling on the catchment was not lost and reached the river channels and then the Caspian Sea (note that the mean annual runoff coefficient in the modern climate for the Volga basin is 0.35, i.e. almost twice as low). As a result, the assessed permafrost-induced changes in the runoff coefficient could themselves lead to an increase in the mean runoff even with a decrease in the mean precipitation comparing with the modern one. And this growth became especially noticeable due to the reduced evaporation from the catchment area caused by the decrease in the air humidity deficit during the post-LGM period and the Oldest Dryas (see Fig. 5). At the same time, the mean runoff visibly dropped during the Bølling period in spite of the permafrost presence that can be explained by a 5-15% decrease in precipitation with a simultaneous 40-45% increase in evaporation (owing to the rise in air humidity deficit) during this period comparing with the previous ones. During the Allerød, the mean runoff was also less that during the post-LGM or the Oldest Dryas, but the difference is not as significant as for the Bølling, owing to the rising precipitation and decreasing evaporation. The response of different parts of the Volga River basin to climate impacts differed from the response of the entire basin as a whole (see Fig. 6).

During the high-flow post-LGM and Oldest Dryas periods, the river runoff was mostly formed in the right-bank sub-catchments of the middle Volga: e.g. within the boundary of the modern Oka River basin, the runoff was 70% more than the spatially averaged one for the Volga basin. This result is confirmed by the data of a paleogeographic reconstruction of the runoff of ancient channels, most of the traces of which are located on the right bank of the middle Volga. On the contrary, on the catchment areas of the Upper Volga and the left-bank part of the middle Volga (Kama basin), the river runoff is estimated to be 30-40% less than the average value for the basin.

According to the simulation results, significant changes occurred in the intra-annual flow regime of the Volga in comparison with the modern regime. In the modern climate, the high flow season runs from April to June and makes up 54% of the annual runoff. In the considered paleo-periods, the high-flow season was a month later (from May to July), and the share of the annual runoff for these months varied from 75% to 85% with the largest value in the Oldest Dryas (Fig. 7). The simulated runoff from the sub-basins of the Oka and Kama Rivers, as well as from the Upper Volga was generally characterized by the same tendencies as for the runoff from the whole Volga. The most notable difference was a significant increase in the Oka freshet during the post-LGM and the Oldest Dryas, which we explained by a larger influence of permafrost together with the increased snow water equivalent due to an increased sum of the solid precipitation as mentioned above. The long-term mean of the annual peak discharge at the outlet of the Volga River during the post-LGM and the Oldest Dryas turned out to be 3 times higher than the corresponding mean simulated under the modern climate, and reached the values of 100,000 m$^3$/s. The mean maximum discharge of the Oka River was as much as 4 times higher than the modern value, reaching 21,000 m$^3$/s during the Oldest Dryas. A significant increase in the mean peak discharge of snowmelt flood compared to the current one was also obtained for the Upper Volga (3.7 times) and for the Kama (2.5 times). Peak flow makes the greatest contribution to the re-shaping of river channels, activates sediment flow and processes of transformation of channel forms. According to the





hypothesis of Sidorchuk et al. (2021), it was the snowmelt floods that turned out to be the main driver of fluvial
activity and the formation of the palaeochannels occurring in the modern Volga basin and formed between 17.5 ka
BP and 14 ka BP.

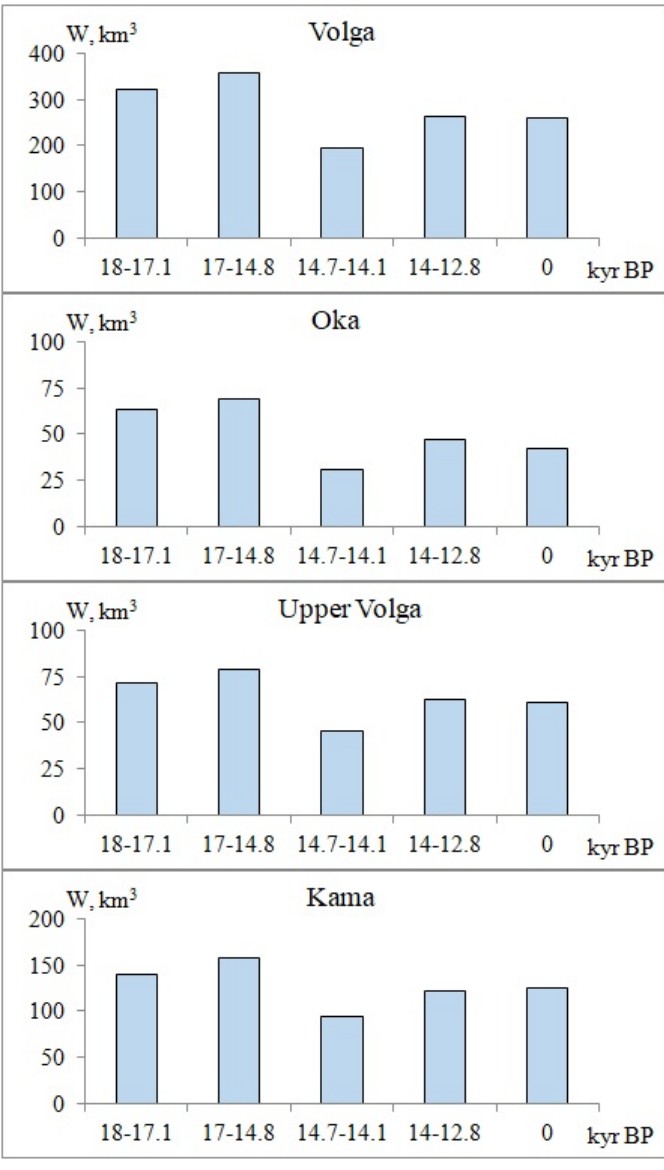

**Figure 6: The mean annual runoff simulated for different periods of deglaciation and for the modern climate. Top to down:**
**the entire Volga basin, the Oka River Basin (right-bank part of the middle Volga), upper part of the basin, the Kama River**
**Basin (left-bank part of the middle Volga).**



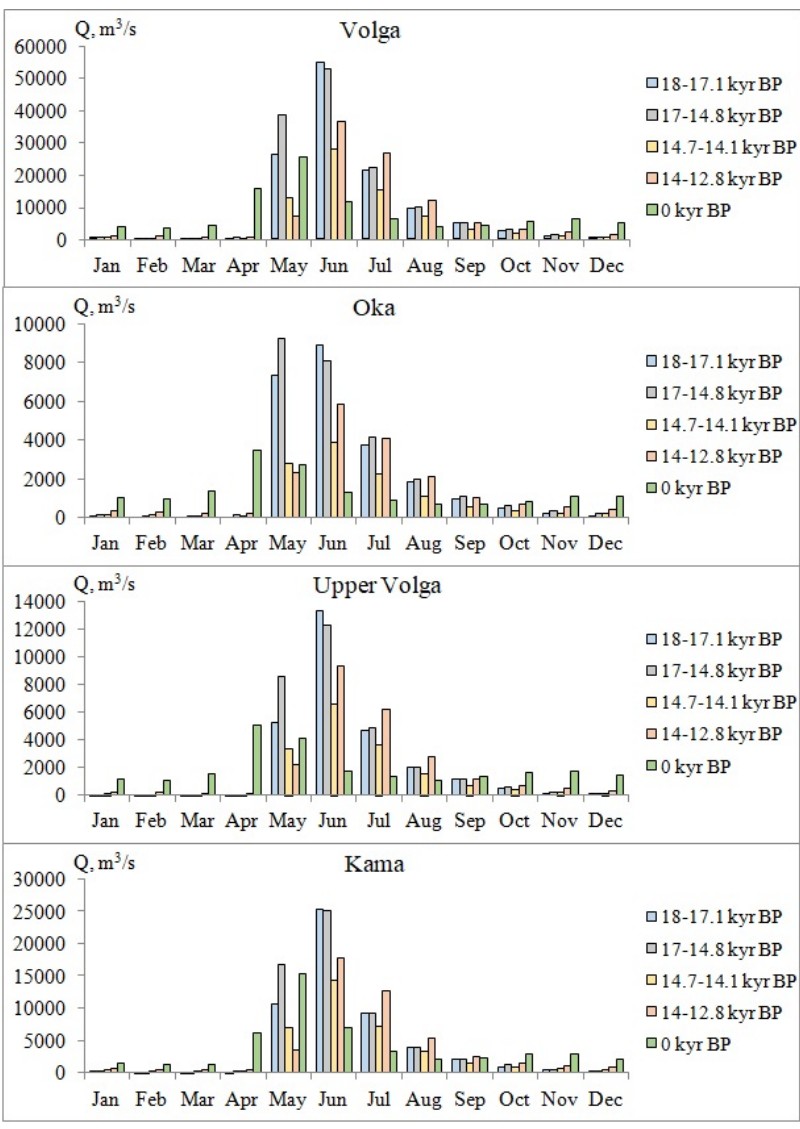

**Figure 7: The mean monthly flow simulated for the different periods of deglaciation and for the modern climate. Top to down: the entire Volga basin, the Oka River Basin (right-bank part of the middle Volga), upper part of the basin, the Kama River Basin (left-bank part of the middle Volga).**

Our results do not contradict this hypothesis, since the largest increase in the simulated mean peak flow occurred in the Volga basin during the Oldest Dryas period (17-14.8 kyr BP).

Thus, we summarized that, according to the data of paleoclimate modeling, the climate of the Volga basin in the period from 18 kyr BP to the end of the Oldest Dryas (14.8 kyr BP) was characterized by low air temperature (11-13°C less than in the modern climate) and low precipitation (24-32% less than in the modern climate). At the same time, according to our experiments with the hydrological model, the mean annual Volga runoff during the Oldest Dryas (17-14.8 kyr BP) could reach up to 360 km$^3$, which is almost 40% higher than the modern runoff, and the mean annual peak flow could increase 3 times. The main factors of the increased runoff were a decrease in evaporation from the



Volga paleo-catchment as well as the spread of permafrost reducing runoff losses due to infiltration into soils, which
all together compensated, over and above, for the decrease in precipitation.
Note that the significant hydrological role of permafrost in the considered paleoperiod could be significantly less in
the process of its degradation in later periods. This can be evidenced, in particular, by the end of increased flow shortly
after 14 ka BP, i.e. in the Allerød, which can hypothetically be associated with thawing of the permafrost by that time.
However, the permafrost completely recovered during the Younger Dryas stadial (12.8-11.8 ka BP), but the formation
of large palaeochannels did not resume during this period. On the contrary, it was noted above that there is a dip in
dates for the 12.5-11.5 ka BP interval, which may indicate a decrease in fluvial activity. This is also supported by the
coincidence of this period with a drop in the sea level, the Yenotaevkian regression (Makshaev and Tkach, 2023).
**5 Conclusions**
Our study was aimed at verifying the physical consistency of the hypothesis asserting the hydroclimatic origin of the
Early Khvalynian transgression of the Caspian Sea. When *a priori* formulating the hypothesis, we firstly relied on the
up-to-date and well-founded OSL-datings (Kurbanov et al., 2021, 2022, 2023; Butuzova et al., 2022; Taratunina et
al., 2022), which referred the sea level stage well above +10 m a.s.l. (likely up to +22 ÷ +35 m a.s.l.) to the final period
of deglaciation, 17-13 kyr BP.  Nowadays, this is the highest dated sea level rise in the Quaternary history of the
Caspian Sea, since the maximum stage of the Early Khvalynian transgression (+48+50 m a.s.l.) has still not been dated
in any geochronological study. Secondly, we relied on the results of recent (Panin et al., 2020, 2021; Borisova et al.,
2021) and earlier (Kalinin et al., 1966; Panin et al., 2005; Sidorchuk et al., 2009) publications, which argued a
negligible contribution of meltwater runoff (due to the Scandinavian ice-sheet melting and outflows of ice-dammed
proglacial lakes) to the transgression of the sea during the considered, 17-13 kyr BP, period. Thirdly, our hypothesis
was based on the ubiquitous presence of large river palaeochannels, whose age was estimated within the close interval,
18-13 kyr BP, in the Caspian Sea catchment and adjacent river basins (Borisova et al., 2006; Sidorchuk et al., 2009;
Panin et al., 2013, 2017; Panin and Matlakhova, 2015). Herewith, the palaeochannels are located in various parts of
the Volga basin, including those completely isolated not only from the last, but also from all Quaternary glaciations,
so the glacial meltwater was unlikely to contribute to their formation (Sidorchuk et al., 2009; 2021).
Thus, previous studies have given us the reasons to believe that the hypothesis put forward does not contradict the
present knowledge on the nature of the Early Khvalynian transgression. That is why we reduced the hypothesis
verification to evaluation of its physical feasibility, i.e. the physical feasibility of the CSL rise above +10 m a.s.l. under
the climate of the deglaciation period, 17-13 kyr BP, in the absence of visible glacial meltwater effect. We carried out
a comprehensive study of the physical consistency of the proposed hypothesis and obtained the following new results:
1. Using the coupled ocean and sea-ice general circulation model INMIO COMPASS – CICE driven by the climate
model INMCM4.8 in accordance with the PMIP4 and CMIP6 modelling protocols, we estimated the equilibrium water
runoff (irrespective of its origin), which could be sufficient to maintain the considered sea level under the modelled
effective evaporation from the entire sea surface area. We found that the mean equilibrium runoff into the Caspian
Sea for its highest dated transgressive state at +35 m a.s.l. (17-13 kyr BP) should fall within the range of 400-470
km$^3$/year. Assuming that the contribution of the Volga River runoff to the total river discharge in that period was close
to the modern one (about 80%), we estimated the river runoff from the Volga River basin during the aforementioned
period as 320-375 km$^3$/year, i.e. 1.3-1.5 times larger than the present day's annual runoff.



2. An extensive 14C-dating of the activity of palaeochannels located in the valleys of 18 rivers in the Volga basin we conducted, allowed us to narrow down the time frames of the epoch of high river discharge to 17.5-14 ka BP and relate the estimate of the annual Volga runoff magnitude derived earlier from the size of the palaeochannels (420 km$^3$/year (Sidorchuk et al., 2021)) to this epoch. Again, the updated time frames are almost identical to the aforementioned modern dating of the main phase of the Early Khvalynian transgression (17-13 ka BP), i.e. the estimates obtained by the independent methods turned out to be very close. Importantly, the estimate of the runoff that formed the studied palaeochannels occurred not far from and higher than the above maximum estimate of the equilibrium runoff: 420 km$^3$/year and 375 km$^3$/year, respectively. That is, the river flow passing through the ancient palaeochannels could maintain the sea level above +10 m a.s.l. under the climate of the considered epoch. As a result, we argued that 17.5-14 ka BP were thousands of years with a huge water inflow capable of maintaining the Caspian Sea level at the maximum dated marks of the Early Khvalynian transgression, and this inflow was not of glacial origin.

3. Using an ECOMAG-based hydrological model of the Volga runoff generation forced by paleoclimate data, we analyzed physically consistent mechanisms of an extraordinary high water inflow into the Caspian Sea both in the absence of visible glacial meltwater effect and under the drier and colder climate than the modern one (e.g., during the Oldest Dryas, 17-14.8 kyr BP, the air temperature was 10.9°C and precipitation was 24% less than in the modern climate). Nevertheless, our numerical experiments demonstrated that the mean annual Volga runoff during the Oldest Dryas could reach up to 360 km$^3$, which is almost 40% higher than the modern runoff, and the mean annual peak flow could increase 3 times. The main factors of the increased runoff were the spread of permafrost which resulted in a sharp drop in infiltration into the frozen ground and reduced evaporation from the Volga paleo-catchment, which all together compensated, over and above, for the decrease in precipitation. A huge growth of peak flow during the Oldest Dryas, 17-14.8 kyr BP, greatly contributed to the processes of river channel transformation and could have formed the giant channels over the ancient Volga catchment.

Thus, our results do not contradict the hypothesis put forward, that the Early Khvalynian transgression of the Caspian Sea could be initiated and maintained solely by hydroclimatic factors within the deglaciation period, 17-13 ka BP. Also, the hypothesis has proven to be physically feasible, since we found a possible cause of the huge inflow into the Caspian Sea in the absence of visible glacial meltwater contribution.

**Code/Data availability**

Paleoclimate Simulation Datasets related to this paper can be found at https://pure.mpg.de/pubman/faces/ViewItemOverviewPage.jsp?itemId=item_3187396_4, an open-source online data repository hosted at MPG PuRe (Kageyama et al., 2021).

**Author contribution**

**Alexander Gelfan:** Conceptualization of the study, Methodology of paleo-hydrological study, Writing, Reviewing and Editing; **Andrey Panin**: Methodology of paleochannels dating, Field works; Writing, Reviewing and Editing; **Andrey Kalugin:** Paleo-hydrological simulations, Writing and Editing; **Polina Morozova**: Paleo-climate simulations, Writing; **Vladimir Semenov**: Methodology of assessing equilibrium river inflow into the sea, Writing; **Alexey Sidorchuk:** Methodology of assessing paleochannel flow; **Vadim Ukraintsev:** Paleochannels dating, Field works; **Konstantin Ushakov:** coupled ocean and sea-ice simulations.



**Competing interests**
The authors declare that they have no known competing financial interests or personal relationships that could have
appeared to influence the work reported in this paper.
**Acknowledgements**
Radiocarbon dating of alluvial deposits and the numerical experiments with the ocean model were financially
supported by the Russian Science Foundation (Grant 19-17-00215). Hydrological simulations were designed within
the framework of the State Assignment theme № FMWZ-2022-0001. Geomorphological investigations in river
floodplain contribute to the State Assignment theme № FMGE-2019-0005.
The present work was carried out within the framework of the Panta Rhei Research Initiative of the International
Association of Hydrological Sciences (IAHS).

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
