# Peer review of "Hydroclimatic processes as the primary drivers of the Early 2 Khvalynian transgression of the Caspian Sea: new developments"

_EGUsphere, 2023_

## Author Comment (AC1)

**Authors' responses to the Reviewer's comments**

We are grateful for reviewing our manuscript and providing positive and useful feedback. The review identifies some unclear issues and helps us to improve presentation of our approach in the revised version of the manuscript. In the following, we are answering all Reviewer's comments one by one.

The Reviewer's comments are in blue.

Our responses are in black.

Added fragments of paper are in *italics*.

The paper covers relevant scientific topics and is very important to understanding the natural processes that took place in the Caspian Region. I consider the paper to be of high value in general, but I would like to address certain comments that would add correctness to the paper.

We are very grateful for the positive evaluation of our work, the careful reading of it, and the recommendations made.

The review of the positions about the age of the Early Khvalynian transgression (lines 45-50) leads to the conclusion that prior to 1990s, there existed belief about the synchronous nature of the Early Khvalynian transgression and the Early Valdai (Early Weichselian, MIS 4) glacial epoch. Then, (referenced Svitoch et al., 1994, 1998), emerged a notion about a "young age" of the transgression based primarily on the radiocarbon data. However, these points of view existed concurrently and were the subjects of fierce debate in scientific community. A.A. Svitoch (whose works are referenced in your paper) defended the position on the "young" age of the transgression even as early as in 1970s. But the scientists who supported the other point of view (G.I. Rychagov, V.A. Zubakov) persisted in their opinion and believed that the radiocarbon method yielded faulty results (and later, the OSL method) and was not suitable for dating of the Lower Khvalynian sediments.

The remark that the opinion about the young age of the Early Khvalynian transgression did not arise in the 1990s but much earlier is fully accepted. We have corrected the corresponding part of the Introduction and added several references (Kaplin et al., 1972, 1973; Svitoch and Parunin, 1973; Svitoch and Yanina, 1983).

> *Before the 1990s, most researchers believed that the maximum phase of the Khvalynian transgression was synchronous to the Early Valdai (Early Weichselian, MIS 4) glaciation of the Russian Plain and occurred 50-70 ka BP (see reviews by Kislov et al., 2014; Arslanov et al., 2016 and references there). Nevertheless, the first radiocarbon ($^{14}C$) dating data allowed already in the early 1970s to formulate the idea of a younger age of this transgression, dating to the very end of the Late Pleistocene (Kaplin et al., 1972, 1973; Svitoch and Parunin, 1973; Svitoch and Yanina, 1983). The accumulation of geochronometric, mostly $^{14}C$ data is increasingly argued in favor of a younger age of the Early Khvalynian transgression, corresponding to the second half of the last glaciation (Late Valdai, Late Weichselian, MIS 2) (Svitoch et al., 1994, 1998; Svitoch and Yanina, 1997).*

Furthermore, the interpretation of the conclusion on the age boundaries of the Khvalynina transgression reached by R. Makshaev and N. Tkach (2023) is not entire correct. R. Makshaev and N. Tkach (2023) presented analyses of not 180 dates, but of 234 (these represent all the radiocarbon dates available to date) and for the entire (!) Caspian. They did not identify two transgressive events in the Early Khalynian post LGM (as you present in the paper), but conjectured on the position of the levels at separate time segments based on the radiocarbon dates.

Indeed, these authors collected a database of 234 dates. By saying that they used 180+ dates, we meant that they plotted their curve of sea level change in Khvalynian time (Fig. 5 in their paper) using 182 dates, since only for these dates elevation data are available. Nevertheless, we agree that

all dates are important for determining the general range of transgression time, and we now give both figures.

The Reviewer points out: "they did not identify two transgressive events in the Early Khalynian post LGM (as you present in the paper), but conjectured on the position of the levels at separate time segments based on the radiocarbon dates". We meant that there are two peaks on the Caspian Sea level change curve constructed by them - in the LGM and in the Late Glacial. But the Reviewer is right that the authors did not distinguish them as separate phases of transgression. that the authors did not distinguish them as separate phases of transgression. In the text, we replaced "transgression phases" with "highstands".

So, we agreed with the Reviewer's criticism and corrected the corresponding text fragment as follows

> *Makshaev and Tkach (2023), based on a generalization of 234 14C dates, of which elevation data were available for 182 dates, attributed the Early Khvalynian stage of the Caspian Sea to the period 36-12.5 ka BP. In their opinion, sea level exceeded the contemporary level at the beginning of MIS 2 (28-25 ka BP). This was followed by two highstands at 25-18 ka BP (level reached +10+15 m a.s.l.) and 17-13.5 ka BP (+20÷+22 m a.s.l.), separated by a sea level drop between 18 and 17 ka BP. These authors date the Yenotayevka regression and the subsequent Late Khvalynian transgression to 12.5-8.5 ka BP.*

Besides, the radiocarbon and OSL dating correlate well and therefore, it doesn't seem to be quite feasible to consider only OSL dating for the Lower Volga Region when building models for the entire Caspian.

We do agree with that, and we can only confirm that we consider the whole massive set of dates when we talk about the chronology of the transgression.

The presented paper does not consider the Manych Strait, which played an important role in the Early Khvalynian basin. Therefore, how the Manych strait affected to the changes of water balance of the Early Khvalynian basin from the post-LGM to the Allerød? In addition, it is necessary to consider the existing radiocarbon and OSL dates obtained from lower Khvalynian deposits of Manych (Svitoch et al., 2010: Pleistocene of the Manych. Questions of the structure and development; Semikolennych et al., 2022: Dating the Khvalynian strait within the late Pleistocene history of the Manych Depression).

We agree that understanding the history of the Caspian Sea levels is impossible without the Manych. We understand that the runoff through the Manych is an important part of the water balance of the Khvalynian Sea, and when it started, the level of the Caspian had to fall and could not rise above the runoff threshold. If we start to talk about the flow along the Manych, we should provide data on the identification of Khvalynian sediments in it and the results of their dating, which are characterized by a large scatter, and we will have to discuss the reliability of these dates. In addition, it will be necessary to give estimates of runoff values for Manych and to explain how they were obtained. In general, this is a rather voluminous conversation and lies away from the purpose of our article, which is not to reconstruct the history of the Caspian Sea levels but to clarify the sources of water that caused the Khvalyn transgression. However, we agree that we cannot completely ignore this part of the Caspian history, and therefore we have added in the Introduction a mention of the flow along the Manych with appropriate references.

> *After the maximum level was reached, there was a breakthrough of the Caspian into the Manych Depression, which caused a westward flow into the Black Sea (Svitoch et al., 2010; Semikolennych et al., 2022).*

**Specific comments:**

Line 56: Yenotaevian regression

Corrected.

Line 563: Yenotaevkian regression

Corrected.

Line 676: Chepalyga, A. L.: Late glacial great flood in the Ponto-Caspian basin, in: The Black Sea Flood Question: Changes in Coastline, Climate, and Human Settlement, edited by: Yanko-Hombach, V., Gilbert, A.S., Panin, A., and Dolukhanov, P.M., Springer, Dordrecht, 119–148, https://doi.org/10.1007/978-1-4020-5302-3_6, 2007.

Among the editors - Nicolae Panin from Romania, not A. Panin

Thank you for the necessary clarification. The reference was changed.

Line 823: Rychagov, G.I.: Late Pleistocene history of the Caspian Sea, in: *Comprehensive* Studies of the Caspian Sea, edited by: Leontiev, O.K., Maev, E.G., MSU Press., Moscow, 18–29, 1974 (in Russian).

Corrected.

Line 855: Svitoch, A. A., and Yanina, T. A. (Eds.): Quaternary Deposits of the Caspian Sea Coasts, MSU Press, Moscow, 267 pp., 1997 (in Russian).

Not entirely clear "(Eds.)." Editors? This is a monograph of two of these authors.

Corrected.

Line 871: Varuschenko, S. I., Varuschenko, A. N., and Klige, R. K. (Eds.): Changes in the Regime of the Caspian Sea and Closed Basins in Paleotime, Nauka, Moscow, 239 pp., 1987 (in Russian).

The same. Not entirely clear "(Eds.)"

Corrected.

And as the conclusion: Thank you for the interesting paper.

We are sincerely grateful for your interest in our research.